# Digital Competence, Use, Actions and Time Dedicated to Digital Devices: Repercussions on the Interpersonal Relationships of Spanish Adolescents

**DOI:** 10.3390/ijerph191610358

**Published:** 2022-08-19

**Authors:** Nieves Gutiérrez Ángel, Isabel Mercader Rubio, Rubén Trigueros Ramos, Nieves Fátima Oropesa Ruiz, Jesús Nicasio García-Sánchez, Judit García Martín

**Affiliations:** 1Departamento de Psicología, Universidad de Almería, 04120 Almería, Spain; 2Departamento de Psicología, Sociología y Filosofía, Universidad de León, 24071 León, Spain; 3Departamento de Psicología Evolutiva y de la Educación, Universidad de Salamanca, 37007 Salamanca, Spain

**Keywords:** digital media, adolescents, PISA 2018, attitude, use of internet

## Abstract

Digital media play a fundamental role in the social, emotional, and cognitive development of adolescents, since they involve a very significant use and investment of time at this age. The objectives of this work are twofold: analyze the use of the Internet and digital devices by Spanish young people outside school, and the time they use them and their attitude towards the use of digital devices, as well as to identify the effects of the use of internet and digital devices on social and interpersonal relationships. The sample is composed of 35,943 students of Compulsory Secondary Education, from different Spanish high schools that participated in the PISA 2018 Report. The data provided by this study confirm the widespread consumption of digital devices. Identified as actions that they carry out every day were: the use of online chat, use of social networks to contact their friends, and surfing the internet for fun. Regarding the attitude towards digital devices, the participants say they feel comfortable using digital devices and discovering new applications or games. However, we also found as one of the most relevant results of this study the fact that participants say they feel bad if they do not have internet connection.

## 1. Introduction

The term digital competence is not a new concept, and its meaning has changed over time [1]. Nowadays, it refers to a set of technical-processual, cognitive, and socioemotional skills needed to live, learn, and work in a digital society [2]. From these ideas, we can establish that digital competence has different dimensions: technological, informational and multimedia, and communicative [3]. These involve reading, writing and arithmetic, and the objective use of technology in personal, social, and work environments, since these literacies cannot be separated from the social and educational needs of the society in which we live [4]. It is precisely this society in which digital media have been postulated as the protagonists that have transformed our lives, both socially and personally [3,5,6,7,8]. Internet is a tool that offers infinite possibilities to adolescents, ranging from communication with family members to the collection of different types of information, social networks, games, and content downloads. Refs. [9,10] state that these have given rise to new opportunities for leisure and entertainment [11] and have revolutionized our mechanisms for thinking, studying, communicating, and playing [12]. Therefore, these digital media play a fundamental role in the social, emotional, and cognitive development of adolescents, as they involve a very significant use and investment of time at this age [13,14].

In terms of their use, in recent years, there has been an increase in the use by adolescents of both digital devices [15] and the Internet [16]. The increase in both the use of the Internet and digital devices has been parallel, a fact that has also been added to the great diversity and types of existing devices, and the ease of access and use of these [17]. This is revealed by the “Survey on equipment and use of information and communication technologies in households”, which puts internet use among the population aged between 16 and 24 at around 98.1% for men and 97.9% for women, where the high figures for the use of these devices can be seen. In general, one of the most frequent uses of the Internet corresponds to interpersonal communication, through social networks, instant messaging, or blogs [17], online games, searching for information, consulting news, and shopping [18].

The most frequent uses of digital devices are mobile phones, video game consoles, and tablets [19]. Despite the benefits and advantages offered by the Internet (such as new ways of learning, communicating, relating, and entertaining oneself), there are several studies that warn of its dangers [20,21], including the fact that excessive time on the Internet and its misuse can be an educational, social, and health problem for adolescents. Current studies have even shown a link between adolescent depression and abusive use of the Internet [19,20], but also between abusive use of the Internet and the negative repercussions it can have on academic performance, or even on relationships with others [19]. It is a social problem, because adolescents may be subject to situations of exclusion or victimization [20]. In addition, it is a health problem, because internet abuse can lead to different mental problems, the most serious of which are suicidal ideation or depression [21,22,23]. Adolescence is a developmental stage characterized by the presence of high vulnerability and susceptibility [20]. In the face of all these issues, there are a large number of studies [24,25,26,27] that have aimed to analyze the use that adolescents make of the Internet.

There are studies about problematic internet use (PIU), which has increased in prevalence due to the pandemic and the virtualization of teaching. By PIU, we mean a major risk factor, both academically and for psychopathological symptoms (i.e., depression, anxiety, and insomnia) in adolescence [21]. Ultimately, PIU refers to a lack of ability on the part of the subject to record internet use leading to feelings of distress and functional impairment [22]. The data are alarming, for example, recent studies in China showed that 17.3% of adolescents under 18 years of age had these behaviors [20], which are closely related to emotional management problems, sleep disorders, or problems in the educational environment. [23]. However, other studies have highlighted that, in addition to these risks, one of the major consequences of IPU is social isolation [24,25,26,27]. In other words, studies have shown a direct and positive correlation between the time adolescents spend online and PIU. In a way, the more time they spend online, the more time they remain disconnected from the real world [28]. In addition, in this regard, other research has highlighted the fact that, compared to adolescents in general, people with PIU experience more social recognition in the online world than offline [29]. In short, we can state that adolescents with PIU, in addition to having problems with depression and anxiety, also experience alterations in their own lifestyle, hygiene and sleep habits, loss of control, anger, symptoms of distress, family conflicts, and social isolation [30,31,32,33]. Therefore, the aims of this study are twofold: to analyze the use that young Spaniards make of the Internet and digital devices outside school, the time they use them, and their attitudes towards the use of digital devices. It also aims to identify the effects of the use of the Internet and digital devices on social and interpersonal relationships.

## 2. Materials and Methods

The method used is correlational, corresponding to an ex post facto, retrospective, and comparative design, since different variables, such as sex, are compared with other types of variables, in this case, dependent variables, which correspond to the use of the Internet, the frequency of such use, and the attitude towards the use of digital devices.

### 2.1. Participants

The sample is composed of 35,943 students of Compulsory Secondary Education from different Spanish high schools that participated in the 2018 PISA Report. The mean age of the sample is 15.83 years, with a standard deviation SD = 0.28. Regarding gender, 50% (*N* = 17,987) are male and 50% (*N* = 17,956) are female. As for nationality, 90.7% (*N* = 31901) were of Spanish nationality and 9.3% (*N* = 3253) were of a nationality other than Spanish. Finally, 82.9% (*N* = 29,129) had not repeated a grade and 11.8% (*N* = 6814) had repeated a grade.

### 2.2. Instrument

The instruments used are the placement tests referring to the 2018 PISA Report, from which only those items referring to questions of a socio-demographic nature and those related to the frequency of ICT use outside of school and internet use, the frequency of such use, and the attitude towards the use of digital devices were chosen. Of the total items in the PISA report, we chose only those related to the time of use of digital devices, the actions carried out through such devices, and the attitudes towards them.

It is the OECD countries that publish a report (International Student Assessment Program, PISA) of statistics to establish the direction of the economy, industry, and education in such countries. In the educational field, they investigate and present information related to mathematical, scientific, and problem-solving content, but they are also investigating various types of student knowledge, such as information and communication technologies (ICT). In addition, it corresponds, therefore, to a large-scale triennial international educational survey, carried out by the Organization for Economic Co-operation and Development (OECD), which assesses the academic performance of 15 year olds every three years. Therefore, it is a scale of categories, widely used in research in the field of psychology to measure preferences, attitudes, or beliefs, through which the sample gives an answer based on a set of specific categories distributed in order of frequency or quantity. In this case, in the Likert scale used, the value of zero corresponded to not agreeing at all, and the value of five to totally agreeing.

### 2.3. Procedure

The results have been collected and processed from the database corresponding to the responses of the PISA 2018 Report level tests. Said database is available on the website of the Ministry of Education and Vocational Training and the INEE (National Institute for Educational Evaluation).

### 2.4. Data Analysis

In this study, the statistical analysis programs SPSS version 26 and AMOS version 22 were used to perform various analyses in order to analyze the relationships between the study variables, such as descriptive and frequency analyses, bivariate correlations, and a model of structural structures. In addition, confidence and descriptive analyses (means and standard deviation) were performed.

For the model of structural structures, the maximum likelihood method and a bootstrapping of 6000 iterations were changed. The fit indices that are taken into account to judge the factorial validity of the proposed model are the following: χ^2^/df, incremental fit index (IFI), comparative fit index (CFI), Tucker Lewis index (TLI), standardized root mean square residual (SRMR), and root mean square error of approximation (RMSEA) plus its confidence interval (CI) at 90%. Thus, the fit indices are: χ^2^/df, values between 2 and 3; IFI, CFI, and TLI, values greater than 0.95; SRMR, values below 0.06; and RMSEA, with values below 0.83.

## 3. Results

### 3.1. Descriptive Uses of Digital Devices

First, participants were asked about the age at which they first used a digital device. Thus, while males stated that it was between 7 and 9 years with 36.8%, females stated that it was between 4 and 6 years with 33.6%.

The age at which they accessed the Internet for the first time was also asked; both males and females reported accessing the Internet for the first time between the ages of 7 and 9 most frequently, with 45% and 43.6%.

We also asked how much time they spend on the Internet, and 30% of both men and women spend between 2 and 4 h.

### 3.2. Time of Use and Actions across Digital Devices

Regarding the actions related to the use of ICT in which the participants spent more time, the results show that some actions are performed to a greater extent than others. Thus, regarding online games, 35.9% never or almost never play single-player games 41.8% use collaborative online games, or 52.4% play online games through social networks.

On the other hand, regarding academic tasks, they never or hardly ever use social networks to communicate with other students about schoolwork, with 41.7%, upload content to the school website, with 44.1%, or consult the school website, with 44.4%.

In terms of the use of certain digital devices, among the tasks that they only recognize as being carried out via mobile devices once or twice a month are downloading an app, with 33.1%, or sending homework and communicating with teachers, with 34.7%. In addition, they acknowledge using digital devices once or twice a week to send emails, with 30.9%, read news on the Internet, with 25.2%, get practical information on the Internet, with 27.6%, surf the Internet to do homework, with 36.3%, or surf the Internet to find explanations for homework, with 29.3%.

On the other hand, they reported daily use of online chatting, with 77.3%, using social networks, with 67.9%, surfing the Internet for fun, with 52.4%, and downloading music, movies, or games, with 26.9%.

### 3.3. Attitudes towards the Use of Digital Devices

In this case, the statements were aimed at finding out participants’ personal tastes and attitudes towards digital devices. Thus, 49.9% of the participants stated that they like to share digital content. In addition, they like to use different digital devices in their day-to-day life according to 53.1%. They also agree that they are totally excited to discover new devices or applications, with 38.6%. In addition, they acknowledge using them to make friends, with 38.1%.

From another point of view, participants were also asked if they felt bad if they did not have an internet connection available. In this case, 43% said they agreed.

This was similar to losing track of time when using digital devices, where 44.4% acknowledged forgetting time when using digital devices.

### 3.4. Influence of the Uses of Digital Devices on Interpersonal Relationships

Descriptive statistics, reliability, and bivariate correlations:

As can be seen in Table 1, the correlations between the study variables were positive, reflecting the reciprocity between the study variables. On the other hand, the reliability analysis through Cronbach’s alpha showed that each of the three factors had scores above 0.70.

### 3.5. Structural Equation Model

The purpose of using an SEM model from a theoretical point of view lies in analyzing the relationships between different variables and their inter-relation (see Table 2). In this way, the SEM model itself and Figure 1 allow us to visualize the relationship that is initially established between the use of digital devices and the use of digital communication devices and, secondly, between the use of digital devices, the use of digital communication devices, and interpersonal relationships.

To explain the relationships between the study variables, a model with a set of predictive relationships was hypothesized (Figure 1). The fit indices revealed that the fits were adequate: χ^2^ (87, *N* = 35,943) = 187.32, χ^2^/df = 2.15, *p* < 0.001, TLI = 0.96, IFI = 0.96, CFI = 0.96, RMSEA = 0.051 (90% CI = 0.048–0.057), and SRMR = 0.038.

The relationships established between the study variables through the SEM are as follows:(a)The correlation between the use of digital devices and the use of digital communication devices was positive (β = 0.67, *p* < 0.001).(b)The use of digital devices showed positive effects on interpersonal relationships (β = 0.41, *p* < 0.001).(c)The use of digital communication devices showed positive effects on interpersonal relationships (β = 0.52, *p* < 0.001).

## 4. Discussion

The main objective or purpose of this study was to analyze the use of the Internet and digital devices by young Spanish people outside of school, taking as variables the time they use these devices and the attitudes they have towards the use of digital devices. In addition, the effects of the use of the Internet and digital devices on face-to-face social relationships established by adolescents were analyzed.

The main findings of this study confirm the widespread use of digital devices in adolescence, with results similar to those obtained in previous studies [34,35,36]. However, one of the novelties established by this study is the identification of the average age of initiation of its use. Specifically, this study places the average age of initiation of its use in the case of men between 7 and 9 years, and, in the case of women, between 4 and 6 years. The study provides similar results for early ages of digital device use [37,38] investigations that established 11 years as the age of onset of use, or even a previous investigation in time that places the age of onset at 7 years [39].

Regarding the time of internet use outside of school, the participants place the average time between 2 and 4 h per day, results that coincide with those reported [15,39].

On the other hand, as actions that adolescents carry out on a daily basis, we find use of online chat, use of social networks to contact their friends, and surf the Internet to have fun, results that are similar to the study of [40]. This is followed by making friends through these means and downloading music, movies, or games. So, we can establish that the use of digital devices has a strong socializing character, hence the explanation of our results regarding the correlation between the use of digital devices and the use of digital communication devices, as well as the fact that the use of digital devices shows positive effects on interpersonal relationships, since, for adolescents, it is a means of relating to their peers.

SEM results have shown that the use of digital devices is positively correlated with the use of digital devices for communication. This result is related to the fact that many of the current advances in the field of new technologies are marked by the increase and quality of devices and software related to the field of communication. In addition, the use of digital and communication devices has been positively related to interpersonal relationships. These relationships are mainly due to the fact that young people make extensive use of these devices, due to their ease of use and accessibility, to interact with their social environment, either directly (for example, telephone call, chat, and video call) or indirectly (e.g., uploading videos or images on social networks).

However, this use of digital devices is not as widespread among the participants in terms of educational use, since the participants acknowledge that they never or almost never use the Internet to download educational applications, consult educational websites, communicate with other students about work from school, upload content to the school website, or check the school website [41].

In terms of attitudes towards digital devices, participants reported feeling comfortable using digital devices and discovering new apps or games. In addition, they consider the Internet to be a great resource for information of interest to them. However, we also found cases in which they admit to feeling bad if they do not have an internet connection or losing track of time during their employment. These results coincide with those reported by [42,43], for whom these excessive attitudes may even correspond to withdrawal symptoms in the face of addiction.

Finally, regarding the influence of these uses on interpersonal relationships, the results obtained show that, in the case of the participants who like to make friends through digital devices, they do not spend any days with friends after school and spend less time away from home with friends. In addition, the feelings they experience when they are out with their friends correspond to a greater extent to boredom. These isolation behaviors related to the preference for digital devices over being with friends have already been highlighted in other studies, such as the one carried out by [13,44], which could even be said to result in social isolation [45,46,47,48,49]. Therefore, we could even be facing cases of PIU.

## 5. Conclusions

All this leads us to ask ourselves, as a future educational implication, if we are really training our students in digital competence from an ethical, civic, and respectful point of view, or, on the contrary, the digital competence of students in the current Spanish educational context is merely technical. As the results of this research show, the educational use of digital devices is less than their recreational use.

The main limitations of this work correspond to the fact that the way of evaluating these elements has corresponded to self-report measures. However, we consider it appropriate to bet on digital competence assessment scales validated in terms of digital literacy itself. The solution paths or future perspectives of the limitations of the present study, in addition to taking these limitations into account, are related to identifying which psychological and educational determinants affect the development of students’ digital competence, as well as their efficiency in achieving academic goals, persistence and resistance to achieving these goals and objectives, the need for belief in one’s own ability or self-efficacy towards digital competence and literacy, involvement or engagement and commitment or commitment to learning, or other relevant psychological variables, which have been shown to be key in the attainment of competence expertise and producing effects on the achievement of academic goals and learning outcomes, including personal and social adjustment.

## Figures and Tables

**Figure 1 ijerph-19-10358-f001:**
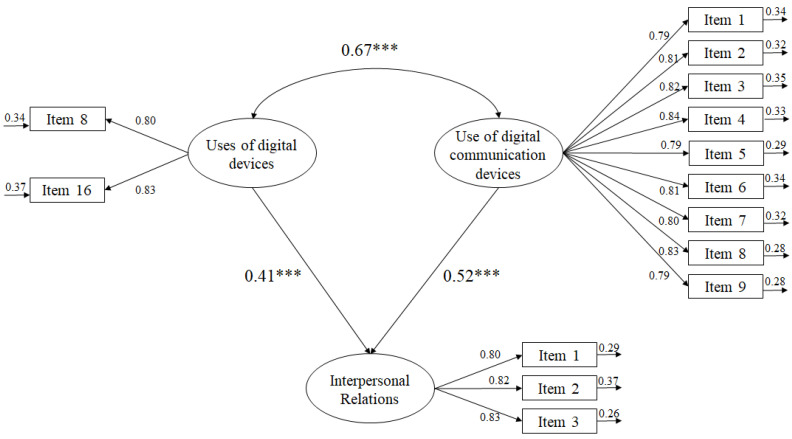
Relationships between the study variables through an SEM. *** *p* < 0.001.

**Table 1 ijerph-19-10358-t001:** Description of the actions related to the use of ICT.

communicate with other students	41.7% (*N* = 1859)
upload content to the school’s website	44.1% (*N* = 12,466)
download an app	33.1% (*N* = 9968)
submit assignments and communicate with teachers	34.7% (*N* = 9898)
send emails	30.9% (*N* = 9363)
read news on the Internet	27.6% (*N* = 8444)
get practical information on the Internet	27.6% (*N* = 8444)
surf the Internet to do homework	36.3% (*N* = 10,565)
browse the Internet to find explanations of duties	29.3% (*N* = 8380)

**Table 2 ijerph-19-10358-t002:** Preliminary analysis.

Factors	M	SD	α	1	2	3
Uses of digital devices	3.34	1.08	0.80	-	0.45 ***	0.48 ***
Use of digital communication devices	3.12	1.10	0.82		-	0.61 ***
Interpersonal relations	3.64	0.88	0.77			-

Note: *** *p* < 0.001; SD = standard deviation; M = mean.

## Data Availability

Not applicable.

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
