# Peer review of "Digital Competence, Use, Actions and Time Dedicated to Digital Devices: Repercussions on the Interpersonal Relationships of Spanish Adolescents"

_ijerph, 2022, doi:10.3390/ijerph191610358_

Round 1

Reviewer 1 Report

The sample size in this article is sufficient so that there is no shortage of directions to analyse or worth analysing, while it will appear to be a very full article, which is a good point. There is no problem with the structure of the article, either.

What needs to be improved is the background section at the beginning, which is too concise and deserves further introduction and analysis of the current situation in Spain

Author Response

  1. a further introduction and analysis of the current situation in Spain.

HAS BEEN COMPLETELY REWRITTEN

Reviewer 2 Report

The main problem of the current version of this article is that this article is poorly written and hard to read. I do not consider myself an expert in English and I do not usually comment on grammar and style at all, but the text looks like a draft (due to numerous typos, unnessesary ellipsises and spaces). See the attached file for more details (please, note that I could not possibly mention all incorrect sentences and only mentioned those that I was sure needed correction). Also, there should be more information anout the methods and the discussion should not only repeat the results, but include more context.

Author Response

  1. The main problem with the current version of this article is that it is poorly written and difficult to read. I don't consider myself an expert in English and I don't usually comment on anything about grammar and style, but the text looks like a draft (due to numerous typos, ellipses, and unnecessary spaces).

THE WORDING OF THE WORK HAS BEEN REVISED

  1. In addition, there should be more information about the methods and the discussion should not only repeat the results, but include more context.

THE DISCUSSION HAS BEEN REMADE

Reviewer 3 Report

I suggest the author re-read the text and write it in a manner that readers will have no problem understanding the paper. In the current form, the paper has too many language and stylistic errors in order for reviewers to focus on the content, and not on, for example:

- Figure in Spanish; no title of the figure:

- ". While it manifests more frequently the answer option almost every day, with actions  172 related to the download of music, movies or games with 26.9% (N = 8134) ."

- " The most frequent  145 responses offered by both men and women tend to spend between 2 and 4 hours a   day  146 connected, with 31.6% (N=5145), and 31.1% (N=4987), respectively."

- "As for the least employed, before which participants say they never or almost never  154 use   more frequently," Not understandable to an English-speaking audience.

- "And he also wondered" Who is "he"?

- "gender or even digitalization.  63 Skills.  64" A one-word sentence?

- "An educational problem due to its negative projection in terms of daily habits and  55 routines, academic grades and even personal relationships [19]." Not understandable to an English-speaking audience.

- "Without forgetting  238 variables such as the time they use such devices and the attitudes they have towards the  239 use of digital devices." Not understandable to an English-speaking audience.

- "All this, without leaving behind the fact of how to identify the  240 effects of the use of the internet and digital devices on the social relationships established  241 by adolescents, taking into account the recent specific literature existing to analyze if the  242 use of digital devices causes isolation with the rest of equals, or on the contrary, it is a tool  243 used by young people as a means of socialization.   244" Not understandable to an English-speaking audience.

- "However, one of the novelties established by this work is to identify the average age of  247 onset in the employment of the same, specifically," Not understandable to an English-speaking audience.

- "Similar results at early ages of digital device use pro- 252 vided by the study [37-38]. "

- " communicate with other students about the school. work, "

- "And  286 therefore, to be in front of cases of PIU."

- "However,first of all this framework, of which at  47 first glance benefits and possibilities of new ways of learning, communicating, relating  48 and entertaining ourselves are supposed [15] there are several studies [16-17] " Not understandable to an English-speaking audience, and probably not even to authors.

Above are just a few examples of carelessly written text. I do wonder why should reviewers even bother with reading the paper (although I did) when it is obvious that the authors did not take the writing seriously?

Author Response

The figure and the full text have been revised and redrafted

Reviewer 4 Report

Thank you to the authors for designing and executing this study.  It is a strong study that I found interesting, relevant to today's world, and of potential interest to IJERPH readers.  Despite the many strengths of the the paper, however, there are a number of soft spots that limit its potential for scientific appeal.  I have outlined the strengths and weaknesses below:

Strengths

(1) Good review of the relevant literature--please consider the American Academy of Pediatrics' position on digital and screen time as appropriate

(2) Good application to epidemiology

(3) Generally strong design, reporting, and scientific writing--but with obvious limitations

Weaknesses

(1) This paper needs someone well-trained in both English and scientific writing to review and improve upon the writing of the paper. The authors have done an admirable job thus far, but there are enough awkwardly worded sentences and paragraphs that stand in the way of its credibility  (see lines 47-52 for a quick example).  My recommendation is for the team to find a skillful proofreader, trained in English, and have him/her smooth out the (many) rough spots.  The authors have a very good message to share with the scientific community, it would be a shame to weaken it with poor grammar and syntax.  

(2) There is an inconsistency between the way the title reads and the the various purpose statements throughout the paper.  The title implies that interpersonal relationships is the primary focus yet throughout the paper, characteristics of digital media use seems to be primary.  My recommendation is to address this inconsistency by sticking to one them or the other.  

(3) In terms of the Methods and Data Analysis sections, I would encourage the team to be much more explicit and consistent with their description of analyses.  As currently written, the text appears to be a mish-mash of references to methods. 

(a) Lines 92-95 undermines their study design.  Their study is a retrospective, quasi-experimental study design that includes a series of descriptive and inferential correlational methods. 

(b) Throughout the paper there are a number of findings that either report correlations or differences and then follow up with a test of significance by gender.  Yet, this comparison of sex differences is not featured in the introduction or the analysis section.  This is huge omission and confuses the reader when s/he gets to the Results and Discussion sections--as to why everything is being viewed through a "gender-based" lens.  This should be addressed in the Introduction and telegraphed clearly in the analysis section--so readers know what to expect and why!

(c) There are references to subject pronouns without any clear references (e.g., 'he' in ln 139, 'they' ln 164). Please be specific about the writing. The authors are encouraged to ask their proofreader to be on the lookout for this type of wording. 

(d) With respect to analyses, specifically, the team should lay out their analyses in the same order in which they appear in the Results section--and then follow that with the same order in the Discussion section.  The reader should not have surprises along the way.  In addition the team should make it clear why different analyses are being conducted--not just that they "were used" (please see lines 120-131). Convince the reader that all analyses not only have a place in your study design but a reason for inclusion, as well. Bottom line:  tighten up the many different analyses so that readers know where and how they fit in. 

(4) The structural equation modeling analysis constitutes a key part of their paper but receives fleeting attention in text.  The authors are strongly encouraged to be more clear about why it is in the paper, what role it plays in their study, and then actually interpret the results for the reader.  As of now, the reader is on his/her own to see what is here: (i) what the items are, (ii) the nature and redundancy of the high correlations, (iii) many of the manifest variables only have the word 'item' while others have Spanish terms, (iv) latent constructs are written in Spanish. Way too confusing for the readers.  The authors are encouraged to present the diagram in English and help the the reader understand all of the findings. 

In closing, I like this paper very much and am glad the authors have brought this topic and findings into the epidemiologic literature.  This topic is quickly gaining momentum internationally and a study from Spain will be helpful and appreciated by the scientific community.  However, the paper is too early in development to be formally considered for the readership of IJERPH. The topic is a great one and the information is likely to be well received by the readership of IJERPH and the scientific community more broadly, but the paper needs to be refined from a scientific as well as a compositional standpoint before I would consider it appropriate for the journal.  Thanks again to the authors for sharing their work with us. I would be delighted to read subsequent submissions if the reviewing editor thinks appropriate.    

Author Response

(1) This document needs someone well trained in both English and scientific writing to review and improve the writing of the document. The authors have done an admirable job so far, but there are enough poorly worded sentences and paragraphs that stand in the way of their credibility (see lines 47-52 for a quick example). My recommendation is that the team find a skilled, English-trained proofreader and that he/she soften the (many) rough spots. The authors have a very good message to share with the scientific community, it would be a shame to weaken it with bad grammar and syntax.

THE WORDING OF THE WORK HAS BEEN REVISED

(2) There is an inconsistency between the way the title is read and the various statements of purpose throughout the document.

The title implies that interpersonal relationships are the main focus, however, throughout the document, the characteristics of the use of digital media seem to be the main ones. My recommendation is to address this inconsistency by sticking to one or the other.

The title has been modified

(3) As for the Methods and Data Analysis sections, I would encourage the team to be much more explicit and consistent with their description of the analyses. As currently written, the text appears to be a hodgepodge of method references.

  1. Lines 92-95 undermine the design of his studio. Their study is a retrospective, quasi-experimental study design that includes a number of descriptive and inferential correlational methods.
  2. Throughout the paper there are a number of findings that report correlations or differences and then follow up with a gender significance test. However, this comparison of sex differences is not presented in the introduction or in the analysis section.

1) This document needs someone well trained in both English and scientific writing to review and improve the writing of the document. The authors have done an admirable job so far, but there are enough poorly worded sentences and paragraphs that stand in the way of their credibility (see lines 47-52 for a quick example). My recommendation is that the team find a skilled, English-trained proofreader and that he/she soften the (many) rough spots. The authors have a very good message to share with the scientific community, it would be a shame to weaken it with bad grammar and syntax.

THE WORDING OF THE WORK HAS BEEN REVISED

(2) There is an inconsistency between the way the title is read and the various statements of purpose throughout the document.

The title implies that interpersonal relationships are the main focus, however, throughout the document, the characteristics of the use of digital media seem to be the main ones. My recommendation is to address this inconsistency by sticking to one or the other.

The title has been modified

(3) As for the Methods and Data Analysis sections, I would encourage the team to be much more explicit and consistent with their description of the analyses. As currently written, the text appears to be a hodgepodge of method references.

  1. Lines 92-95 undermine the design of his studio. Their study is a retrospective, quasi-experimental study design that includes a number of descriptive and inferential correlational methods.
  2. Throughout the paper there are a number of findings that report correlations or differences and then follow up with a gender significance test. However, this comparison of sex differences is not presented in the introduction or in the analysis section.

Round 2

Reviewer 2 Report

In my opinion, the language has improved significantly in this version, even though not all of my suggestions were implemented. However, this highlighted some other possible improvements from the content point of view.

1) Results, section "Descriptive Uses of digital devices": Chi-Square test shows statistically significant differences according to sex for all three described variables: first use of a digital device, age of first Internet access and how much timespend on the Internet. The problem here is that only one result for each group is described (I have mentioned it in the previous review and this seems to be ignored). Specifically, the article says that:

A) 36.8% of male participants first used digital devices at the age 7-9 while 33.6% of female participants first used digital devices at the age 4-6.  Chi-Square test shows statistically significant differences so we can ASSUME (but we cannot be sure about this, because we do not have any information about the rest 63.2% of male participants and 66.4% of female participants) that women were found to start their digital devices use earlier than men. 

B) 45% of men and 43.6% of women reported having first accessed the Internet at the age of 7-9. HOWEVER, Chi-Square test showed statistically significant differences and this time we cannot deduce what kind of differences are there, as the presented data seems quite similar. So, there might be some significant differences between men and women somewhere in the missing part of the data (the rest 55% of men and 56.4% of women). 

C) The exact same situation happens with the question about hours of Internet use - if there are significant differences between men and women, it should be explicitly stated what are those differences. The easiest way to back up this statement is to provide full percentages for all possible answers, and not only the most frequent answers in each group.

2) How many items there were in each scale mentioned in Table 1? The item number, if it is low, can lower Cronbach's Alpha as well, so the interpretation of the reliability analysis would wary based on the number of items in a scale. Also, Cronbach's Alpha is still referred to as Alpha de Cronbach in the Table 1 footing.

3) What is the theory behind the model (Figure 1)? It is statistically fitting, but what does it mean from a theoretical point of view? For now, everything that you say about the SEM results could be stated based on correlations from Table 1 instead, so I cannot see, why did you need advanced statistics just to state that the scales correlate with each other... 

By the way, if there are really only two items in Uses of digital devices scale and three - in Interpersonal relations scale, it is very much ok for the have Cronbach Alpha levels of 0.8 and 0.77 respectively. 

4) The conclusions from the discussion section are somewhat not fully supported by this article. E.g. "Specifically, this study places the average age of starting to use them in the case of males, between 7 and 9 years old, and in the case of females, between 4 and 6 years old. Similar results for early ages of use of digital devices are provided by the study. Therefore, we can say that this average age is getting lower and lower, and therefore, the starting age is getting younger and younger." - Can you make a direct comparison with some older or newer data to prove that in the text (so that the reader did not have to consult the references articles)?

5) What are the limitations of your study? Please, include this section in the DIscussion.

Altoghether, the collected data is significant and interesting, and I believe it is possible to make a good article out of this data with the authors` skills in statistical analisys. However, in my opinion some more theoretical explanations are needed to finish this work, as for now it only revolves around the statistical procedures performed...

Author Response

Please see the response in the file.

Reviewer 3 Report

The resubmitted paper addresses the question of adolescent media use, which is an important and timely topic.

Let me first point out that in a response to the reviewer, the author only stated one (!) sentence. They did not bother to explain what changes they made to the resubmitted version, nor did they use “Track changes”, so the reviewer would see the changes they made. The reviewers had to find the changes on their own.

However, there are numerous shortcomings in the resubmitted paper that a journal reader will likely notice. Just a few examples:

- numerous sentences have poor structure. In the abstract, one sentence begins with “As well as to identify the effects of the use of internet and digital devices on social and  16 interpersonal relationships”, where “as well” is not an appropriate sentence starter;

- similarly: “Identifying as ac- 19 tions that they carry out every day:”, where “Identifying as” is not an appropriate sentence starter;

- there are many unnecessary fillers in sentences in the main text, for example; “we can consider” (28), “From these ideas we can establish” (30-31), “To this fact can be added the existence of the” (37), “And in relation to these ideas, it can be said that” (47), “we should not forget” (58), “Well, we must not forget” (68), “It is precisely here where we can talk about” (73), etc.

Unfortunately, these examples are not a question of English proofreading but more of a poor judgment on the structure of sentences in scientific writing. I suggest the authors follow the language, expressions and grammatical rules used in scientific journals in English in their future writing efforts.

The first paragraph in the Method section has been criticized in the first version of the paper. It is unclear how (or whether) authors have changed the method section.

The “Instrument” is not described; there is no place in the paper where items are stated. This reduces the chance of reproducibility of other scientists to carry out the analyses, which is one of the tenets of scientific research.

“Data analysis” section does not state why (neither stating the reason nor reference) the described analyses were performed and why specific fit indices were used. In addition, it is not stated why “structural equation model” was used and how this benefits compared to correlational or multivariate analyses.

“Results” section lacks tables which would show the results described in the first few paragraphs of the Results section.

Next, only % need to be stated in the text, not Ns.

After a title of a section of the paper, the first sentence begins with “In this case,” which makes it incomprehensible.

The Discussion section again contains sentences that contain unneeded starters, for example: “Not forgetting variables such as” (223), “And, the fact” (246), “And they” (263), “the  The”, (265), “As well as the” (267), “And therefore,” (277), etc.

One sentence in the Discussion states: “to analyse whether the use of digital  227 devices causes isolation with the rest of peers, or on the contrary, it is a tool used by young  228 people as a means of socialisation.” Unfortunately, it is impossible to carry out an empirical analysis that would confirm either “isolation” or “socialisation”. All media entail socialisation, as media is, including by several definitions, a “socialisation agent”.

Authors also state that “Therefore, we can say that this average age is getting lower and lower, and therefore,  236 the starting age is getting younger and younger”, while they do not provide any concrete numbers (age in years) in their study (or from other studies) that would show a decrease in years (although the decrease might be expected, a reader need concrete numbers and author too in order to make such claims).

A reader wonders how this finding can be interpreted “The SEM results have shown that the use of digital devices is positively correlated  248 with the use of digital devices for communication.”, other that it is the only possible finding to have. Digital devices are mainly used for communication, this is known from other studies. It is a channel for communication, so surely they are positively linked to communication use. This is not a novel finding.

Section “6. Patents” also does not make sense. First, what “patents”?  Second, in “Author Contributions: “, it only states “investigation”, where all author initials are stated. There is no information on “writing the original draft”, “formal analyses” etc. Why the authors did not even bother with filling out the online questionnaire is unclear.

In the “Informed Consent Statement” it is stated “Not applicable.” It is unclear how it is not applicable when human data was used.

This also goes for “Data Availability Statement: Not applicable.”

Finally, let me point out that the Discussion section also lacks a paragraph on the Limitations of the study. That shows a lack of objectivity and self-reflectivity by the authors.

Considering some of the shortcomings of the paper, it is not publishable in IJERPH, in my opinion.

(Then authors revised paper and provided the reviewer point-by-point response,  the reviewer wrote the 3rd review report as follows:)

I appreciate the fact that the authors wrote a point-by-point response.

I have read the paper, and the main comment still stand in the present (third version) of the paper. For example,  the paragraph from my second review still holds:

" - there are many unnecessary fillers in sentences in the main text, for example; “we can consider” (28), “From these ideas we can establish” (30-31), “To this fact can be added the existence of the” (37), “And in relation to these ideas, it can be said that” (47), “we should not forget” (58), “Well, we must not forget” (68), “It is precisely here where we can talk about” (73), etc."

Or, for example, this paragraph from the 3rd version of the paper (see underlined and in bold):

"The main objective or purpose of this study was to analyze the use of the Internet  and digital devices by young Spanish people outside of school, taking as variables the  time they use these devices and the attitudes they have towards the use of digital devices. 
. And the effects of the use of the Internet and digital devices on face-to-face social rela- tionships established by adolescents were analyzed.  "

Most importantly, I don't see the paper satisfactory filling a gap in the literature. It uses a new(er) dataset, but this is not enough for a publication in a top-tier journal, in my view. In the Discussion section, the authors state:

"On the other hand, as actions that adolescents carry out on a daily basis, we find: use  the online chat, use social networks to contact their friends and surf the Internet to have  fun, results similar to the study of [40]."

This is not enough to warrant a scientific publication. There is no novelty in such a finding. I therefore cannot recommend publishing it. (regardless of my view that it seems that the reviewers made more effort with reading the paper (and pointing out numerous issues), than the the authors did writing (or reading) it).

Hopefully, my reviews of this paper were helpful to the Editorial team, regardless of their decision regarding the paper.

Author Response

Dear Reviewer, Thank you very much for your contributions which we take into account to improve

Reviewer 4 Report

The authors have responded to most of my concerns.  As someone who actually writes in this area, I like the paper very much and think it will make an important contribution to the literature.  There are still typos, even in the abstract (sentence 3), and throughout.  To keep the authors and journal from being viewed in a less than optimal light, I would encourage the authors to do another pass through the paper to catch the dependent clauses that are masquerading as independent clauses.  I would also encourage them to make sure to more closely align their statistical analysis statements with the order in which the information is being presented in the Results and Discussion.  As currently written, I do not think the variations interfere with the reading or interpretation.  Thanks for sharing the revised document with me.  

Author Response

REVIEWER 3. ROUND 2

The authors have responded to most of my concerns. As someone who actually writes in this area, I really like the article and I think it will make an important contribution to literature. There are still typos, even in the summary (sentence 3), and everywhere. To prevent authors and the journal from looking less than optimal, I would recommend authors to do another review of the paper to detect dependent clauses that masquerade as stand-alone clauses.

The entire document and the expressions used have been revised

I would also encourage them to ensure that they more closely align their statistical analysis statements with the order in which the information is presented in the Results and Discussion. As currently written, I don't think variations interfere with reading or interpretation. Thank you for sharing the revised document with me.

This section has been restructured following the order in which the information is presented in the Results and discussion (lines 240-252)

This manuscript is a resubmission of an earlier submission. The following is a list of the peer review reports and author responses from that submission.